# LeRaC: Learning Rate Curriculum

## Abstract

Most curriculum learning methods require an approach to sort the data samples by difficulty, which is often cumbersome to perform. In this work, we propose a novel curriculum learning approach termed **Le**arning **Ra**te Curriculum (LeRaC), which leverages the use of a different learning rate for each layer of a neural network to create a data-free curriculum during the initial training epochs. More specifically, LeRaC assigns higher learning rates to neural layers closer to the input, gradually decreasing the learning rates as the layers are placed farther away from the input. The learning rates increase at various paces during the first training iterations, until they all reach the same value. From this point on, the neural model is trained as usual. This creates a model-level curriculum learning strategy that does not require sorting the examples by difficulty and is compatible with any neural network, generating higher performance levels regardless of the architecture. We conduct comprehensive experiments on 10 data sets from the computer vision (CIFAR-10, CIFAR-100, Tiny ImageNet, ImageNet-200, PASCAL VOC), language (BoolQ, QNLI, RTE) and audio (ESC-50, CREMA-D) domains, considering various convolutional (ResNet-18, Wide-ResNet-50, DenseNet-121, YOLOv5), recurrent (LSTM) and transformer (CvT, BERT, SepTr) architectures. We compare our approach with the conventional training regime, as well as with Curriculum by Smoothing (CBS), a state-of-the-art data-free curriculum learning approach. Unlike CBS, our performance improvements over the standard training regime are consistent across all data sets and models. Furthermore, we significantly surpass CBS in terms of training time (there is no additional cost over the standard training regime for LeRaC). Our code is freely available at: http//github.com/link.hidden.for.review.

## 1 Introduction

Curriculum learning (Bengio et al., 2009) refers to efficiently training effective neural networks by mimicking how humans learn, from easy to hard. As originally introduced by Bengio et al. (2009), curriculum learning is a training procedure that first organizes the examples in their increasing order of difficulty, then starts the training of the neural network on the easiest examples, gradually adding increasingly more difficult examples along the way, until all training examples are fed to the network. The success of the approach relies in avoiding imposing the learning of very difficult examples right from the beginning, instead guiding the model on the right path through the imposed curriculum. This type of curriculum is later referred to as data-level curriculum learning (Soviany et al., 2022). Indeed, Soviany et al. (2022) identified several types of curriculum learning approaches in the literature, dividing them into four categories based on the components involved in the definition of machine learning given by Mitchell (1997). The four categories are: data-level curriculum (examples are presented from easy to hard), model-level curriculum (the modeling capacity of the network is gradually increased), task-level curriculum (the complexity of the learning task is increased during training), objective-level curriculum (the model optimizes towards an increasingly more complex objective). While data-level curriculum is the most natural and direct way to employ curriculum learning, its main disadvantage is that it requires a way to determine the difficulty of data samples. Despite having many successful applications (Soviany et al., 2022; Wang et al., 2022), there is no universal way to determine the difficulty of the data samples, making the data-level curriculum less applicable to scenarios where the difficulty is hard to estimate, *e.g.* classification of radar signals. The task-level and objective-level curriculum learning strategies suffer from similar issues, *e.g.* it is hard to create a curriculum when the model has to learn an easy task (binary classification) or the objective function is already convex.

Considering the above observations, we recognize the potential of model-level curriculum learning strategies of being applicable across a wider range of domains and tasks. To date, there are only a

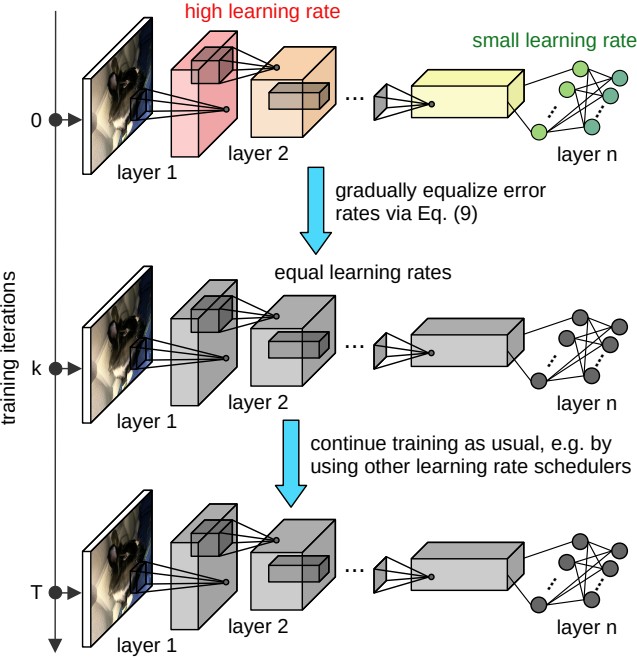

Figure 1: Training based on Learning Rate Curriculum.

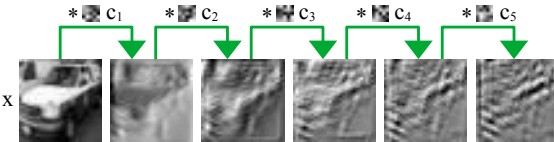

Figure 2: Convolving an image of a car with random noise filters progressively increases the level of noise in the features. A theoretical proof is given in the supplementary.

few works (Burduja & Ionescu, 2021; Karras et al., 2018; Sinha et al., 2020) in the category of pure model-level curriculum learning methods. However, these methods have some drawbacks caused by their domain-dependent or architecture-specific design. To benefit from the full potential of the model-level curriculum learning category, we propose LeRaC (**Le**arning **Ra**te **C**urriculum), a novel and simple curriculum learning approach which leverages the use of a different learning rate for each layer of a neural network to create a data-free curriculum during the initial training epochs. More specifically, LeRaC assigns higher learning rates to neural layers closer to the input, gradually decreasing the learning rates as the layers are placed farther away from the input. This reduces the propagation of noise caused by the multiplication operations inside the network, a phenomenon that is more prevalent when the weights are randomly initialized. The learning rates increase at various paces during the first training iterations, until they all reach the same value, as illustrated in Figure 1. From this point on, the neural model is trained as usual. This creates a model-level curriculum learning strategy that is applicable to any domain and compatible with any neural network, generating higher performance levels regardless of the architecture, without adding any extra training time. To the best of our knowledge, we are the first to employ a different learning rate per layer to achieve the same effect as conventional (data-level) curriculum learning.

As hinted above, the underlying hypothesis that justifies the use of LeRaC is that the level of noise grows from one neural layer to the next, especially when the input is multiplied with randomly initialized weights having low signal-to-noise ratios. We briefly illustrate this phenomenon through an example. Suppose an image $x$ is successively convolved with a set of random filters $c_1$, $c_2$, ..., $c_n$. Since the filters are uncorrelated, each filter distorts the image in a different way, degrading the information in $x$ with each convolution. The information in $x$ is gradually replaced by noise (see Fig. 2), *i.e.* the signal-to-noise ratio increases with each layer. Optimizing the filter $c_n$ to learn a pattern from the image convolved with $c_1$, $c_2$, ..., $c_{n-1}$ is suboptimal, because the filter $c_n$ will adapt to the noisy (biased) activation map induced by filters $c_1$, $c_2$, ..., $c_{n-1}$. This suggests that earlier filters need to be optimized sooner to reduce the level of noise of the activation map passed to layer

$n$. In general, this phenomenon becomes more obvious as the layers get deeper, since the number of multiplication operations grows along the way. Hence, in the initial training stages, it makes sense to use gradually lower learning rates, as the layers get father away from the input. Our hypothesis is theoretically supported by Theorem 1, and empirically validated in the supplementary.

We conduct comprehensive experiments on 10 data sets from the computer vision (CIFAR-10 (Krizhevsky, 2009), CIFAR-100 (Krizhevsky, 2009), Tiny ImageNet (Russakovsky et al., 2015), ImageNet-200 (Russakovsky et al., 2015), PASCAL VOC (Everingham et al., 2010)), language (BoolQ (Clark et al., 2019), QNLI (Wang et al., 2019), RTE (Wang et al., 2019)) and audio (ESC-50 (Piczak, 2015), CREMA-D (Cao et al., 2014)) domains, considering various convolutional (ResNet-18 (He et al., 2016), Wide-ResNet-50 (Zagoruyko & Komodakis, 2016), DenseNet-121 (Huang et al., 2017), YOLOv5 (Jocher et al., 2022)), recurrent (LSTM (Hochreiter & Schmidhuber, 1997)) and transformer (CvT (Wu et al., 2021), BERT (Devlin et al., 2019), SepTr (Ristea et al., 2022)) architectures. We compare our approach with the conventional training regime and Curriculum by Smoothing (CBS) (Sinha et al., 2020), our closest competitor. Unlike CBS, our performance improvements over the standard training regime are consistent across all data sets and models. Furthermore, we significantly surpass CBS in terms of training time, since there is no additional cost over the conventional training regime for LeRaC, whereas CBS adds Gaussian smoothing layers.

In summary, our contribution is threefold:

- We propose a novel and simple model-level curriculum learning strategy that creates a curriculum by updating the weights of each neural layer with a different learning rate, considering higher learning rates for the low-level feature layers and lower learning rates for the high-level feature layers.
- We empirically demonstrate the applicability to multiple domains (image, audio and text), the compatibility to several neural network architectures (convolutional neural networks, recurrent neural networks and transformers), and the time efficiency (no extra training time added) of LeRaC through a comprehensive set of experiments.
- We demonstrate our underlying hypothesis stating that the level of noise increases from one neural layer to another, both theoretically and empirically.

## 2 RELATED WORK

Curriculum learning was initially introduced by Bengio et al. (2009) as a training strategy that helps machine learning models to generalize better when the training examples are presented in the ascending order of their difficulty. Extensive surveys on curriculum learning methods, including the most recent advancements on the topic, were conducted by Soviany et al. (2022) and Wang et al. (2022). In the former survey, Soviany et al. (2022) emphasized that curriculum learning is not only applied at the data level, but also with respect to the other components involved in a machine learning approach, namely at the model level, the task level and the objective (loss) level. Regardless of the component on which curriculum learning is applied, the technique has demonstrated its effectiveness on a broad range of machine learning tasks, from computer vision (Bengio et al., 2009; Gui et al., 2017; Jiang et al., 2018; Shi & Ferrari, 2016; Soviany et al., 2021; Chen & Gupta, 2015; Sinha et al., 2020) to natural language processing (Platanios et al., 2019; Kocmi & Bojar, 2017; Spitkovsky et al., 2009; Liu et al., 2018; Bengio et al., 2009) and audio processing (Ranjan & Hansen, 2018; Amodei et al., 2016).

The main challenge for the methods that build the curriculum at the data level is measuring the difficulty of the data samples, which is required to order the samples from easy to hard. Most studies have addressed the problem with human input (Pentina et al., 2015; Jiménez-Sánchez et al., 2019; Wei et al., 2021) or metrics based on domain-specific heuristics. For instance, the text length (Kocmi & Bojar, 2017; Cirik et al., 2016; Tay et al., 2019; Zhang et al., 2021) and the word frequency (Bengio et al., 2009; Liu et al., 2018) have been employed in natural language processing. In computer vision, the samples containing fewer and larger objects have been considered to be easier in some works (Soviany et al., 2021; Shi & Ferrari, 2016). Other solutions employed difficulty estimators (Ionescu et al., 2016) or even the confidence level of the predictions made by the neural network (Gong et al., 2016; Hacohen & Weinshall, 2019) to approximate the complexity of the data samples. Such solutions have shown their utility in specific application domains. Nonetheless, measuring the difficulty remains problematic when implementing standard (data-level) curriculum learning strategies, at least in some application domains. Therefore, several alternatives have emerged over time,

handling the drawback and improving the conventional curriculum learning approach. In Kumar et al. (2010), the authors introduced self-paced learning to evaluate the learning progress when selecting training samples. The method was successfully employed in multiple settings (Kumar et al., 2010; Gong et al., 2019; Fan et al., 2017; Li et al., 2016; Zhou et al., 2018; Jiang et al., 2015; Ristea & Ionescu, 2021). Furthermore, some studies combined self-paced learning with the traditional pre-computed difficulty metrics (Jiang et al., 2015; Ma et al., 2017). An additional advancement related to self-paced learning is the approach called self-paced learning with diversity (Jiang et al., 2014). The authors demonstrated that enforcing a certain level of variety among the selected examples can improve the final performance. Another set of methods that bypass the need for predefined difficulty metrics is known as teacher-student curriculum learning (Zhang et al., 2019; Wu et al., 2018). In this setting, a teacher network learns a curriculum to supervise a student neural network.

Closer to our work, a few methods (Karras et al., 2018; Sinha et al., 2020; Burduja & Ionescu, 2021) proposed to apply curriculum learning at the model level, by gradually increasing the learning capacity (complexity) of the neural architecture. Such curriculum learning strategies do not need to know the difficulty of the data samples, thus having a great potential to be useful in a broad range of tasks. For example, Karras et al. (2018) proposed to gradually add layers to generative adversarial networks during training, while increasing the resolution of the input images at the same time. They are thus able to generate realistic high-resolution images. However, their approach is not applicable to every domain, since there is no notion of resolution for some input data types, *e.g.* text. Sinha et al. (2020) presented a strategy that blurs the activation maps of the convolutional layers using Gaussian kernel layers, reducing the noisy information caused by the network initialization. The blur level is progressively reduced to zero by decreasing the standard deviation of the Gaussian kernels. With this mechanism, they obtain a training procedure that allows the neural network to see simple information at the start of the process and more intricate details towards the end. Curriculum by Smoothing (CBS) (Sinha et al., 2020) was only shown to be useful for convolutional architectures applied in the image domain. Although we found that CBS is applicable to transformers by blurring the tokens, it is not necessarily applicable to any neural architecture, *e.g.* standard feed-forward neural networks. As an alternative to CBS, Burduja & Ionescu (2021) proposed to apply the same smoothing process on the input image instead of the activation maps. The method was applied with success in medical image alignment. However, this approach is not applicable to natural language input, as it is not clear how to apply the blurring operation on the input text.

Different from Burduja & Ionescu (2021) and Karras et al. (2018), our approach is applicable to various domains, including but not limited to natural language processing, as demonstrated throughout our experiments. To the best of our knowledge, the only competing model-level curriculum method which is applicable to various domains is CBS (Sinha et al., 2020). Unlike CBS, LeRaC does not introduce new operations, such as smoothing with Gaussian kernels, during training. As such, our approach does not increase the training time with respect to the conventional training regime, as later shown in the experiments included in the supplementary. In summary, we consider that the simplicity of our approach comes with many important advantages: applicability to any domain and task, compatibility with any neural network architecture, time efficiency (adds no extra training time). We support all these claims through the comprehensive experiments presented in Section 4.

In the supplementary, we explain how LeRaC is different from learning rate schedulers and optimizers. We also present additional experiments to support our claims.

## 3 METHOD

Deep neural networks are commonly trained on a set of labeled data samples denoted as:

$$S = \{(x_i, y_i) | x_i \in X, y_i \in Y, \forall i \in \{1, 2, ..., m\}\}, \tag{1}$$

where $m$ is the number of examples, $x_i$ is a data sample and $y_i$ is the associated label. The training process of a neural network $f$ with parameters $\theta$ consists of minimizing some objective (loss) function $\mathcal{L}$ that quantifies the differences between the ground-truth labels and the predictions of the model $f$:

$$\min_{\theta} \frac{1}{m} \sum_{i=1}^{m} \mathcal{L}\left(y_i, f(x_i, \theta)\right). \tag{2}$$

The optimization is generally performed by some variant of Stochastic Gradient Descent (SGD), where the gradients are back-propagated from the neural layers closer to the output towards the

neural layers closer to input through the chain rule. Let $f_1$, $f_2$, ...., $f_n$ and $\theta_1$, $\theta_2$, ..., $\theta_n$ denote the neural layers and the corresponding weights of the model $f$, such that the weights $\theta_j$ belong to the layer $f_j$, $\forall j \in \{1, 2, ..., n\}$. The output of the neural network for some training data sample $x_i \in X$ is formally computed as follows:

$$\hat{y}_i = f(x_i, \theta) = f_n \left( ...f_2 \left( f_1 \left( x_i, \theta_1 \right), \theta_2 \right) ...., \theta_n \right). \tag{3}$$

To optimize the model via SGD, the weights are updated as follows:

$$\theta_j^{(t+1)} = \theta_j^{(t)} - \eta^{(t)} \cdot \frac{\partial \mathcal{L}}{\partial \theta_j^{(t)}}, \forall j \in \{1, 2, ..., n\}, \tag{4}$$

where $t$ is the index of the current training iteration, $\eta^{(t)} > 0$ is the learning rate at iteration $t$, and the gradient of $\mathcal{L}$ with respect to $\theta_j^{(t)}$ is computed via the chain rule. Before starting the training process, the weights $\theta_j^{(0)}$ are commonly initialized with random values, *e.g.* using Glorot initialization (Glorot & Bengio, 2010).

Sinha et al. (2020) suggested that the random initialization of the weights produces a large amount of noise in the information propagated through the neural model during the early training iterations, which can negatively impact the learning process. Due to the feed-forward processing that involves several multiplication operations, we argue that the noise level grows with each neural layer, from $f_j$ to $f_{j+1}$. This statement is confirmed by the following theorem:

**Theorem 1.** *Let $s_1 = u_1 + z_1$ and $s_2 = u_2 + z_2$ be two signals, where $u_1$ and $u_2$ are the clean components, and $z_1$ and $z_2$ are the noise components. The signal-to-noise ratio of the product between the two signals is lower than the signal-to-noise ratios of the two signals,* i.e.:

$$\mathrm{SNR}(s_1 \cdot s_2) \leq \mathrm{SNR}(s_i), \forall i \in \{1, 2\}. \tag{5}$$

*Proof.* A theoretical proof is given in the supplementary. □

The same issue can occur if the weights are pre-trained on a distinct task, where the misalignment of the weights with a new task is likely higher for the high-level (specialized) feature layers. To alleviate this problem, we propose to introduce a curriculum learning strategy that assigns a different learning rate $\eta_j$ to each layer $f_j$, as follows:

$$\theta_j^{(t+1)} = \theta_j^{(t)} - \eta_j^{(t)} \cdot \frac{\partial \mathcal{L}}{\partial \theta_j^{(t)}}, \forall j \in \{1, 2, ..., n\}, \tag{6}$$

such that:

$$\eta^{(0)} \geq \eta_1^{(0)} \geq \eta_2^{(0)} \geq ... \geq \eta_n^{(0)}, \tag{7}$$

$$\eta^{(k)} = \eta_1^{(k)} = \eta_2^{(k)} = ... = \eta_n^{(k)}, \tag{8}$$

where $\eta_j^{(0)}$ are the initial learning rates and $\eta_j^{(k)}$ are the updated learning rates at iteration $k$. The condition formulated in Eq. (7) indicates that the initial learning rate $\eta_j^{(0)}$ of a neural layer $f_j$ gets lower as the level of the respective neural layer becomes higher (farther away from the input). With each training iteration $t \leq k$, the learning rates are gradually increased, until they become equal, according to Eq. (8). Thus, our curriculum learning strategy is only applied during the early training iterations, where the noise caused by the misfit (randomly initialized or pre-trained) weights is most prevalent. Hence, $k$ is a hyperparameter of LeRaC that is usually adjusted such that $k \ll T$, where $T$ is the total number of training iterations.

At this point, various schedulers can be used to increase each learning rate $\eta_j$ from iteration 0 to iteration $k$. We empirically observed that an exponential scheduler is a better option than linear or logarithmic schedulers. We thus propose to employ the exponential scheduler, which is based on the following rule:

$$\eta_j^{(l)} = \eta_j^{(0)} \cdot c^{\frac{l}{k} \cdot \left( \log_c \eta_j^{(k)} - \log_c \eta_j^{(0)} \right)}, \forall l \in \{0, 1, ..., k\}. \tag{9}$$

We set $c = 10$ in Eq. (9) across all our experiments. In practice, we obtain optimal results by initializing the lowest learning rate $\eta_n^{(0)}$ with a value that is around five or six orders of magnitude lower than $\eta^{(0)}$, while the highest learning rate $\eta_1^{(0)}$ is always equal to $\eta^{(0)}$. Apart from these general practical notes, the exact LeRaC configuration for each neural architecture is established by tuning its two hyperparameters $(k, \eta_n^{(0)})$ on the available validation sets.

| Model | Optimizer | Mini-batch | #Epochs | $\eta^{(0)}$ | CBS | | | LeRaC | |
| | | | | | $\sigma$ | $d$ | $u$ | $k$ | $\eta_1^{(0)}$ - $\eta_n^{(0)}$ |
|---|---|---|---|---|---|---|---|---|---|
| ResNet-18 | SGD | 64 | 100-200 | $10^{-1}$ | 1 | 0.9 | 2-5 | 5-7 | $10^{-1}$ - $10^{-8}$ |
| Wide-ResNet-50 | SGD | 64 | 100-200 | $10^{-1}$ | 1 | 0.9 | 2-5 | 5-7 | $10^{-1}$ - $10^{-8}$ |
| CvT-13 | AdaMax | 64-128 | 150-200 | $2 \cdot 10^{-3}$ | 1 | 0.9 | 2-5 | 2-5 | $2 \cdot 10^{-3}$ - $2 \cdot 10^{-8}$ |
| CvT-13$_{\text{pre-trained}}$ | AdaMax | 64-128 | 25 | $5 \cdot 10^{-4}$ | 1 | 0.9 | 2-5 | 3-6 | $5 \cdot 10^{-4}$ - $5 \cdot 10^{-10}$ |
| YOLOv5$_{\text{pre-trained}}$ | SGD | 16 | 100 | $10^{-2}$ | 1 | 0.9 | 2 | 3 | $10^{-2}$ - $10^{-5}$ |
| BERT$_{\text{large-uncased}}$ | AdaMax | 10 | 7-25 | $5 \cdot 10^{-5}$ | 1 | 0.9 | 1 | 3 | $5 \cdot 10^{-5}$ - $5 \cdot 10^{-8}$ |
| LSTM | AdamW | 256-512 | 25-70 | $10^{-3}$ | 1 | 0.9 | 2 | 3-4 | $10^{-3}$ - $10^{-7}$ |
| SepTR | Adam | 2 | 50 | $10^{-4}$ | 0.8 | 0.9 | 1-3 | 2-5 | $10^{-4}$ - $10^{-8}$ |
| DenseNet-121 | Adam | 64 | 50 | $10^{-4}$ | 0.8 | 0.9 | 1-3 | 2-5 | $10^{-4}$ - $5 \cdot 10^{-8}$ |

Table 1: Optimal hyperparameter settings for the various neural architectures used in our experiments. Notice that $\eta_1^{(0)}$ is always equal to $\eta^{(0)}$, being set without tuning. This means that LeRaC has only two tunable hyperparameters, $k$ and $\eta_n^{(0)}$, while CBS (Sinha et al., 2020) has three.

We underline that the output feature maps of a layer $j$ are affected $(i)$ by the misfit weights $\theta_j^{(0)}$ of the respective layer, and $(ii)$ by the input feature maps, which are in turn affected by the misfit weights of the previous layers $\theta_1^{(0)}, ..., \theta_{j-1}^{(0)}$. Hence, the noise affecting the feature maps increases with each layer processing the feature maps, being multiplied with the weights from each layer along the way. Our curriculum learning strategy imposes the training of the earlier layers at a faster pace, transforming the noisy weights into discriminative patterns. As noise from the earlier layer weights is eliminated, we train the later layers at faster and faster paces, until all learning rates become equal at epoch $k$.

From a technical point of view, we note that our approach can also be regarded as a way to guide the optimization, which we see as an alternative to loss function smoothing. The link between curriculum learning and loss smoothing is discussed by Soviany et al. (2022), who suggest that curriculum learning strategies induce a smoothing of the loss function, where the smoothing is higher during the early training iterations (simplifying the optimization) and lower to non-existent during the late training iterations (restoring the complexity of the loss function). LeRaC is aimed at producing a similar effect, but in a softer manner by dampening the importance of optimizing the weights of high-level layers in the early training iterations. Additionally, we empirically observe (see results in the supplementary) that LeRaC tends to balance the training pace of low-level and high-level features, while the conventional regime seems to update the high-level layers at a faster pace. This could provide an additional intuitive explanation of why our method works better.

## 4 EXPERIMENTS

### 4.1 EXPERIMENTAL SETUP

**Data sets.** We perform experiments on 10 benchmarks: CIFAR-10 (Krizhevsky, 2009), CIFAR-100 (Krizhevsky, 2009), Tiny ImageNet (Russakovsky et al., 2015), ImageNet-200 (Russakovsky et al., 2015), PASCAL VOC 2007+2012 (Everingham et al., 2010), BoolQ (Clark et al., 2019), QNLI (Wang et al., 2019), RTE (Wang et al., 2019), CREMA-D (Cao et al., 2014), and ESC-50 (Piczak, 2015). We adopt the official data splits for the 10 benchmarks considered in our experiments. When a validation set is not available, we keep $10\%$ of the training data for validation. Additional details about the data sets are provided in the supplementary.

**Architectures.** To demonstrate the compatibility of LeRaC with multiple neural architectures, we select several convolutional, recurrent and transformer models. As representative convolutional neural networks (CNNs), we opt for ResNet-18 (He et al., 2016), Wide-ResNet-50 (Zagoruyko & Komodakis, 2016) and DenseNet-121 (Huang et al., 2017). For the object detection experiments on PASCAL VOC, we use the YOLOv5s (Jocher et al., 2022) model based on the CSPDarknet53 (Wang et al., 2020) backbone, which is pre-trained on the MS COCO data set (Lin et al., 2014). As representative transformers, we consider CvT-13 (Wu et al., 2021), BERT$_{\text{uncased-large}}$ (Devlin et al., 2019) and SepTr (Ristea et al., 2022). For CvT, we consider both pre-trained and randomly initialized versions. We use an uncased large pre-trained version of BERT. As Ristea et al. (2022), we train SepTr from scratch. In addition, we employ a long short-term memory (LSTM) network (Hochreiter &

| Model | Training Regime | CIFAR-10 | CIFAR-100 | Tiny ImageNet | ImageNet-200 |
|---|---|---|---|---|---|
| ResNet-18 | conventional | $89.20_{\pm 0.43}$ | $71.70_{\pm 0.06}$ | $57.41_{\pm 0.05}$ | $71.66_{\pm 0.10}$ |
| | CBS | $89.53_{\pm 0.22}$ | $\mathbf{72.80}_{\pm 0.18}$ | $55.49_{\pm 0.20}$ | $72.51_{\pm 0.15}$ |
| | LeRaC (ours) | $\mathbf{89.56}_{\pm 0.16}$ | $72.72_{\pm 0.12}$ | $\mathbf{57.86}_{\pm 0.20}$ | $\mathbf{73.17}_{\pm 0.15}$ |
| Wide-ResNet-50 | conventional | $91.22_{\pm 0.24}$ | $68.14_{\pm 0.16}$ | $55.97_{\pm 0.30}$ | $72.83_{\pm 0.13}$ |
| | CBS | $89.05_{\pm 1.00}$ | $65.73_{\pm 0.36}$ | $48.30_{\pm 1.53}$ | $74.75_{\pm 0.08}$ |
| | LeRaC (ours) | $\mathbf{91.58}_{\pm 0.16}$ | $\mathbf{69.38}_{\pm 0.26}$ | $\mathbf{56.48}_{\pm 0.60}$ | $\mathbf{74.88}_{\pm 0.15}$ |
| CvT-13 | conventional | $71.84_{\pm 0.37}$ | $41.87_{\pm 0.16}$ | $33.38_{\pm 0.27}$ | $70.68_{\pm 0.17}$ |
| | CBS | $72.64_{\pm 0.29}$ | $\mathbf{44.48}_{\pm 0.40}$ | $33.56_{\pm 0.36}$ | $69.91_{\pm 0.10}$ |
| | LeRaC (ours) | $\mathbf{72.90}_{\pm 0.28}$ | $43.46_{\pm 0.18}$ | $\mathbf{33.95}_{\pm 0.28}$ | $\mathbf{71.21}_{\pm 0.14}$ |
| CvT-13$_{\text{pre-trained}}$ | conventional | $93.56_{\pm 0.05}$ | $77.80_{\pm 0.16}$ | $70.71_{\pm 0.35}$ | - |
| | CBS | $85.85_{\pm 0.15}$ | $62.35_{\pm 0.48}$ | $68.41_{\pm 0.13}$ | - |
| | LeRaC (ours) | $\mathbf{94.15}_{\pm 0.03}$ | $\mathbf{78.93}_{\pm 0.05}$ | $\mathbf{71.34}_{\pm 0.08}$ | - |

Table 2: Average accuracy rates (in %) over 5 runs on CIFAR-10, CIFAR-100, Tiny ImageNet and ImageNet-200 for various neural models based on different training regimes: conventional, CBS (Sinha et al., 2020) and LeRaC. The accuracy of the best training regime in each experiment is highlighted in bold.

Schmidhuber, 1997) to represent recurrent neural networks (RNNs). The recurrent neural network contains two LSTM layers, each having a hidden dimension of 256 components. These layers are preceded by one embedding layer with the embedding size set to 128 elements. The output of the last recurrent layer is passed to a classifier composed of two fully connected layers. The LSTM is activated by rectified linear units (ReLU). We apply the aforementioned models on distinct input data types, considering the intended application domain of each model. Hence, ResNet-18, Wide-ResNet-50, CvT and YOLOv5 are applied on images, BERT and LSTM are applied on text, and SepTr and DenseNet-121 are applied on audio.

**Baselines.** We compare LeRaC with two baselines: the conventional training regime (which uses early stopping and reduces the learning rate on plateau) and the state-of-the-art Curriculum by Smoothing (Sinha et al., 2020). For CBS, we use the official code released by Sinha et al. (2020) at `https://github.com/pairlab/CBS`, to ensure the reproducibility of their method in our experimental settings, which include a more diverse selection of input data types and neural architectures.

**Hyperparameter tuning.** We tune all hyperparameters on the validation set of each benchmark. In Table 1, we present the optimal hyperparameters chosen for each architecture. In addition to the standard parameters of the training process, we report the parameters that are specific for the CBS (Sinha et al., 2020) and LeRaC strategies. In the case of CBS, $\sigma$ denotes the standard deviation of the Gaussian kernel, $d$ is the decay rate for $\sigma$, and $u$ is the decay step. Regarding the parameters of LeRaC, $k$ represents the number of iterations used in Eq. (9), and $\eta_1^{(0)}$ and $\eta_n^{(0)}$ are the initial learning rates for the first and last layers of the architecture, respectively. We set $\eta_1^{(0)} = \eta^{(0)}$ and $c = 10$ in all experiments, without tuning. In addition, the intermediate learning rates $\eta_j^{(0)}, \forall j \in \{2, 3, ..., n-1\}$, are automatically set to be equally distanced between $\eta_1^{(0)}$ and $\eta_n^{(0)}$. Moreover, $\eta_j^{(k)} = \eta^{(0)}$, *i.e.* the initial learning rates of LeRaC converge to the original learning rate set for the conventional training regime. All models are trained with early stopping and the learning rate is reduced by a factor of 10 when the loss reaches a plateau. Except for the pre-trained models, the weights of all models are initialized with Glorot initialization (Glorot & Bengio, 2010).

**Evaluation.** For the classification tasks, we evaluate all models in terms of the accuracy rate. For the object detection task, we employ the mean Average Precision (mAP) at an intersection over union (IoU) threshold of 0.5. We repeat the training process of each model for 5 times and report the average performance and the standard deviation.

## 4.2 RESULTS

**Image classification.** In Table 2, we present the image classification results on CIFAR-10, CIFAR-100, Tiny ImageNet and ImageNet-200. Since CvT-13 is pre-trained on ImageNet, it does not make sense to fine-tune it on ImageNet-200. Thus, the respective results are not reported. On the one hand, there are two scenarios (ResNet-18 on CIFAR-100, and CvT-13 on CIFAR-100) in which CBS provides the largest improvements over the conventional regime, surpassing LeRaC in the respective

| Training Regime | conventional | CBS | LeRaC (ours) |
|---|---|---|---|
| mAP | $0.832_{\pm 0.006}$ | $0.829_{\pm 0.003}$ | $\mathbf{0.846_{\pm 0.004}}$ |

Table 3: Object detection results of YOLOv5s on PASCAL VOC. The best mAP is in bold.

| Training Regime | Text | | | | Audio | | |
|---|---|---|---|---|---|---|---|
| | Model | BoolQ | RTE | QNLI | Model | CREMA-D | ESC-50 |
| conventional | | $74.12_{\pm 0.32}$ | $74.48_{\pm 1.36}$ | $92.13_{\pm 0.08}$ | | $70.47_{\pm 0.67}$ | $91.13_{\pm 0.33}$ |
| CBS | $BERT_{large}$ | $74.37_{\pm 1.11}$ | $74.97_{\pm 1.96}$ | $91.47_{\pm 0.22}$ | SepTr | $69.98_{\pm 0.71}$ | $91.15_{\pm 0.41}$ |
| LeRaC (ours) | | $\mathbf{75.55_{\pm 0.66}}$ | $\mathbf{75.81_{\pm 0.29}}$ | $\mathbf{92.45_{\pm 0.13}}$ | | $\mathbf{70.95_{\pm 0.56}}$ | $\mathbf{91.58_{\pm 0.28}}$ |
| conventional | | $64.40_{\pm 1.37}$ | $54.12_{\pm 1.60}$ | $59.42_{\pm 0.36}$ | | $67.21_{\pm 0.12}$ | $88.91_{\pm 0.11}$ |
| CBS | LSTM | $64.75_{\pm 1.54}$ | $54.03_{\pm 0.45}$ | $59.89_{\pm 0.38}$ | DenseNet-121 | $68.16_{\pm 0.19}$ | $88.76_{\pm 0.17}$ |
| LeRaC (ours) | | $\mathbf{65.80_{\pm 0.33}}$ | $\mathbf{55.71_{\pm 1.04}}$ | $\mathbf{59.98_{\pm 0.34}}$ | | $\mathbf{68.99_{\pm 0.08}}$ | $\mathbf{90.02_{\pm 0.10}}$ |

Table 4: Left side: average accuracy rates (in %) over 5 runs on BoolQ, RTE and QNLI for BERT and LSTM. Right side: average accuracy rates (in %) over 5 runs on CREMA-D and ESC-50 for SepTr and DenseNet-121. In both domains (text and audio), the comparison is between different training regimes: conventional, CBS (Sinha et al., 2020) and LeRaC. The accuracy of the best training regime in each experiment is highlighted in bold.

| Model | Training Regime | CIFAR-10 | CIFAR-100 | Tiny ImageNet |
|---|---|---|---|---|
| | conventional | $71.84_{\pm 0.37}$ | $41.87_{\pm 0.16}$ | $33.38_{\pm 0.27}$ |
| CvT-13 | LeRaC (logarithmic update) | $\mathbf{72.14_{\pm 0.13}}$ | $\mathbf{43.37_{\pm 0.20}}$ | $\mathbf{33.82_{\pm 0.15}}$ |
| | LeRaC (linear update) | $\mathbf{72.49_{\pm 0.27}}$ | $\mathbf{43.39_{\pm 0.14}}$ | $\mathbf{33.86_{\pm 0.07}}$ |
| | LeRaC (exponential update) | $\mathbf{72.90_{\pm 0.28}}$ | $\mathbf{43.46_{\pm 0.18}}$ | $\mathbf{33.95_{\pm 0.28}}$ |

Table 5: Average accuracy rates (in %) over 5 runs on CIFAR-10, CIFAR-100 and Tiny ImageNet for CvT-13 based on different training regimes: conventional, LeRaC with logarithmic update, LeRaC with linear update, and LeRaC with exponential update (proposed). The accuracy rates surpassing the baseline training regime are highlighted in bold.

cases. On the other hand, there are eight scenarios where CBS degrades the accuracy with respect to the standard training regime. This shows that the improvements attained by CBS are inconsistent across models and data sets. Unlike CBS, our strategy surpasses the baseline regime in all fifteen cases, thus being more consistent. In seven of these cases, the accuracy gains of LeRaC are higher than $1\%$. Moreover, LeRaC outperforms CBS in thirteen out of fifteen cases. We thus consider that LeRaC can be regarded as a better choice than CBS, bringing consistent performance gains.

**Object detection.** In Table 3, we include the object detection results of YOLOv5 (Jocher et al., 2022) based on different training regimes on PASCAL VOC (Everingham et al., 2010). LeRaC exhibits a superior mAP score, significantly surpassing the other training regimes.

**Text classification.** In Table 4 (left side), we report the text classification results on BoolQ, RTE and QNLI. Here, there are only two cases (BERT on QNLI and LSTM on RTE) where CBS leads to performance drops compared with the conventional training regime. In all other cases, the improvements of CBS are below $0.6\%$. Just as in the image classification experiments, LeRaC brings accuracy gains for each and every model and data set. In four out of six scenarios, the accuracy gains yielded by LeRaC are higher than $1.3\%$. Once again, LeRaC proves to be the most consistent regime, generally surpassing CBS by significant margins.

**Speech classification.** In Table 4 (right side), we present the results obtained on the audio data sets, namely CREMA-D and ESC-50. We observe that the CBS strategy obtains lower results compared with the baseline in two cases (SepTr on CREMA-D and DenseNet-121 on ESC-50), while our method provides superior results for each and every case. By applying LeRaC on SepTr, we set a new state-of-the-art accuracy level ($70.95\%$) on the CREMA-D audio modality, surpassing the previous state-of-the-art value attained by Ristea et al. (2022) with SepTr alone. When applied on DenseNet-121, LeRaC brings performance improvements higher than $1\%$, the highest improvement ($1.78\%$) over the baseline being attained on CREMA-D.

### 4.3 ABLATION STUDY

**Comparing different schedulers.** We first aim to establish if the exponential learning rate scheduler proposed in Eq. (9) is a good choice. To test this out, we select the CvT-13 model and change the LeRaC regime to use linear or logarithmic updates of the learning rates. The corresponding results

| Training Regime | $\eta_1^{(0)}$-$\eta_n^{(0)}$ | ResNet-18 | Wide-ResNet-50 |
|---|---|---|---|
| conventional | $10^{-1}$-$10^{-1}$ | $71.70_{\pm 0.06}$ | $68.14_{\pm 0.16}$ |
| LeRaC (ours) | $10^{-1}$-$10^{-6}$ | $\mathbf{72.48}_{\pm 0.10}$ | $\mathbf{68.64}_{\pm 0.52}$ |
| | $10^{-1}$-$10^{-7}$ | $\mathbf{72.52}_{\pm 0.17}$ | $\mathbf{69.25}_{\pm 0.37}$ |
| | $10^{-1}$-$10^{-8}$ | $\mathbf{72.72}_{\pm 0.12}$ | $\mathbf{69.38}_{\pm 0.26}$ |
| | $10^{-1}$-$10^{-9}$ | $\mathbf{72.29}_{\pm 0.38}$ | $\mathbf{69.26}_{\pm 0.27}$ |
| | $10^{-1}$-$10^{-10}$ | $\mathbf{72.45}_{\pm 0.25}$ | $\mathbf{69.66}_{\pm 0.34}$ |
| | $10^{-2}$-$10^{-8}$ | $\mathbf{72.41}_{\pm 0.08}$ | $\mathbf{68.51}_{\pm 0.52}$ |
| | $10^{-3}$-$10^{-8}$ | $\mathbf{72.08}_{\pm 0.19}$ | $\mathbf{68.71}_{\pm 0.47}$ |

Table 6: Average accuracy rates (in %) over 5 runs for ResNet-18 and Wide-ResNet-50 on CIFAR-100 based on different ranges for the initial learning rates. The accuracy rates surpassing the baseline training regime are highlighted in bold.

| Training Regime | $k$ | ResNet-18 | Wide-ResNet-50 |
|---|---|---|---|
| conventional | - | $71.70_{\pm 0.06}$ | $68.14_{\pm 0.16}$ |
| LeRaC (ours) | 5 | $\mathbf{73.04}_{\pm 0.09}$ | $\mathbf{68.86}_{\pm 0.76}$ |
| | 6 | $\mathbf{72.87}_{\pm 0.07}$ | $\mathbf{69.78}_{\pm 0.16}$ |
| | 7 | $\mathbf{72.72}_{\pm 0.12}$ | $\mathbf{69.38}_{\pm 0.26}$ |
| | 8 | $\mathbf{73.50}_{\pm 0.16}$ | $\mathbf{69.30}_{\pm 0.18}$ |
| | 9 | $\mathbf{73.29}_{\pm 0.28}$ | $\mathbf{68.94}_{\pm 0.30}$ |

Table 7: Average accuracy rates (in %) over 5 runs for ResNet-18 and Wide-ResNet-50 on CIFAR-100 using the LeRaC regime until iteration $k$, while varying $k$. The accuracy rates surpassing the baseline training regime are highlighted in bold.

are shown in Table 5. We observe that both alternative schedulers obtain performance gains, but our exponential learning rate scheduler brings higher gains on all three data sets. We thus conclude that the update rule defined in Eq. (9) is a sound option.

**Varying value ranges for initial learning rates.** All our hyperparameters are either fixed without tuning or tuned on the validation data. In this ablation experiment, we present results with LeRaC using multiple ranges for $\eta_1^{(0)}$ and $\eta_n^{(0)}$ to demonstrate that LeRaC is sufficiently stable with respect to suboptimal hyperparameter choices. We carry out experiments with ResNet-18 and Wide-ResNet-50 on CIFAR-100. We report the corresponding results in Table 6. We observe that all hyperparameter configurations lead to surpassing the baseline regime. This indicates that LeRaC can bring performance gains even outside the optimal learning rate bounds, demonstrating low sensitivity to suboptimal hyperparameter tuning.

**Varying k.** In Table 7, we present additional results with ResNet-18 and Wide-ResNet-50 on CIFAR-100, considering various values for $k$ (the last iteration for our training regime). We observe that all configurations surpass the baselines on CIFAR-100. Moreover, we observe that the optimal values for $k$ ($k = 7$ for ResNet-18 and $k = 7$ for Wide-ResNet-50) obtained on the validation set are not the values producing the best results on the test set.

**Summary.** Notably, our ablation results show that the majority of hyperparameter configurations tested for LeRaC lead to outperforming the conventional regime, demonstrating the stability of LeRaC. We present additional ablation results in the supplementary.

## 5 CONCLUSION

In this paper, we introduced a novel model-level curriculum learning approach that is based on starting the training process with increasingly lower learning rates per layer, as the layers get closer to the output. We conducted comprehensive experiments on 10 data sets from three domains (image, text and audio), considering multiple neural architectures (CNNs, RNNs and transformers), to compare our novel training regime (LeRaC) with a state-of-the-art regime (CBS (Sinha et al., 2020)), as well as the conventional training regime (based on early stopping and reduce on plateau). The empirical results demonstrate that LeRaC is significantly more consistent than CBS, perhaps being one of the most versatile curriculum learning strategy to date, due to its compatibility with multiple neural models and its usefulness across different domains. Remarkably, all these benefits come for free, *i.e.* LeRaC does not add any extra time over the conventional approach.

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
