# OpenReview forum: "LeRaC: Learning Rate Curriculum"
_ICLR.cc/2024/Conference — Submitted to ICLR 2024_

### Official Review · Reviewer_sPFc · 2023-10-15

**Soundness:** 3 good
**Presentation:** 3 good
**Contribution:** 3 good
**Rating:** 6
**Confidence:** 2

**Summary:**

This paper introduces a novel approach to curriculum learning in deep neural networks. Curriculum learning is a technique where the learning process is guided by the order in which training samples are presented, typically starting with easier examples and progressively moving to more difficult ones. Traditional curriculum learning methods require sorting data by difficulty, which can be cumbersome. In contrast, LeRaC proposes a data-free curriculum learning strategy that dynamically adjusts the learning rates for different layers of a neural network during initial training epochs. Specifically, it assigns higher learning rates to layers closer to the input, gradually reducing them as layers move away from the input. The learning rates converge to a uniform value, and the model is then trained as usual. This approach is tested across various domains (computer vision, language, and audio) and architectures, outperforming traditional training and a state-of-the-art data-free curriculum learning approach called Curriculum by Smoothing.

**Strengths:**

- This paper is written in a clear and easily comprehensible manner, making it easy for readers to follow.
- LeRaC introduces a unique approach to curriculum learning by dynamically adjusting learning rates for different layers. This eliminates the need for sorting data by difficulty and simplifies the training process.

**Weaknesses:**

see Question.

**Questions:**

- In Figure 2, the authors present a straightforward example illustrating the relationship between shallow and deep features. This example is intuitive and easy to grasp; however, there are some points open for discussion. For instance, the statement "as the information in $x$ is lost" might not necessarily hold when utilizing random convolutional kernels or random transformations, as seen in the popular diffusion model's noise injection process. Therefore, I suggest rephrasing this part.

- Regarding the example mentioned earlier, I believe the network's representation should be considered holistically. Is it meaningful to discuss the representations of individual layers separately? Will merely increasing the learning rate for shallow layer parameters lead to faster convergence of shallow network parameters? The first theorem seems insufficient to address this issue.

- In Equation (9), the author mentions, "we empirically observed that an exponential scheduler is better." It would be beneficial for the author to provide an insightful explanation as to why the exponential scheduler is superior to linear or logarithmic schedulers. Are there any other potentially better scheduling methods, such as cosine learning rate variations?

- The author should provide clarification on the aforementioned points. Once these issues are addressed, I will assess these clarifications in conjunction with feedback from other reviewers to determine whether I should reconsider my evaluation.

---

> ### Author Response · Authors · 2023-11-21
> **Rebuttal letter**
>
> - _In Figure 2, the authors present a straightforward example illustrating the relationship between shallow and deep features. This example is intuitive and easy to grasp; however, there are some points open for discussion. For instance, the statement "as the information in x is lost" might not necessarily hold when utilizing random convolutional kernels or random transformations, as seen in the popular diffusion model's noise injection process. Therefore, I suggest rephrasing this part._
>
> **Reply:** We agree. We wanted to express the fact the signal-to-noise ratio drops from one layer to the other. In the revised manuscript, we rephrased the statement accordingly.
>
> - _Regarding the example mentioned earlier, I believe the network's representation should be considered holistically. Is it meaningful to discuss the representations of individual layers separately? Will merely increasing the learning rate for shallow layer parameters lead to faster convergence of shallow network parameters? The first theorem seems insufficient to address this issue._
>
> **Reply:** In principle, a larger learning rate implies a larger update. However, we do agree that if the learning rate is too high, the model can actually diverge. This is because the gradient describes the loss function in the vicinity of the current location, providing no guarantee for the value of the loss outside this vicinity. Our implementation actually takes this into account. Instead of increasing the learning rate for earlier layers, it reduces the learning rate for the deeper layer to avoid divergence. More precisely, we set the learning rate for the first layer $\eta_1^{(0)}$ to the original learning rate $\eta^{(0)}$ and the other initial learning rates are gradually reduced with each layer i. During training, the lower learning rates are gradually increased, until epoch k. Hence, LeRaC actually slows down the learning for deeper layers, until the earlier layers have learned “good enough” features. We clarified this point in the revised supplementary.
>
> - _In Equation (9), the author mentions, "we empirically observed that an exponential scheduler is better." It would be beneficial for the author to provide an insightful explanation as to why the exponential scheduler is superior to linear or logarithmic schedulers. Are there any other potentially better scheduling methods, such as cosine learning rate variations?_
>
> **Reply:** We believe that a suitable scheduler is one that adjusts the learning rate at layer i proportionally to the estimated signal-to-noise drop at layer i. To demonstrate this, we have plotted the average SNR / entropy of randomly initialized LeNet layers for CIFAR-100 images. The new plot shown in Figure 3 added to the revised supplementary shows that the SNR / entropy drops exponentially, from one layer to the next. This can explain why the exponential scheduler is a more suitable choice. We thank the reviewer for this comment, which lead us to discovering a better explanation for our design choices.

---

> > ### Comment · Reviewer_sPFc · 2023-11-23
> > **Thank you for your response.**
> >
> > Thank you for your detailed response and I tend to keep my score.

---

### Official Review · Reviewer_RRAE · 2023-10-30

**Soundness:** 3 good
**Presentation:** 3 good
**Contribution:** 3 good
**Rating:** 5
**Confidence:** 3

**Summary:**

The paper proposes to gradually change learning rate (LR) for each layer of a neural network during optimization iterations. This "data-free" curriculum learning scheme is based on the noisy amplification by cascaded neural net architecture including CNN, RNN and transformers. The proposed method has been evaluated with many datasets across modalities -- images, text and audio.

**Strengths:**

- The proposed method is based on a good intuition of noise amplified when the layer is close to semantic information.
- Simple idea that performs quite well.

**Weaknesses:**

- Marginal empirical gain. As shown in the Table 2 and 3, most of the gain over CBS, which is the direct competitors, is less than 1 or 2%.
- Method is too simple without intuitive ground that it should work better than others. Although the analysis is intuitively sensible, the simplicity of the method brings marginal performance gain over the direct competitor CBS, even with the quite thorough ablation study with different range of values.
  - As authors mentioned, the empirically chosen exponential based method may not be the best choice or the intuition of noise amplification may not be a serious problem. Given the results, it is difficult to judge the main reason for the unsatisfactory performance.

**Questions:**

- Can you elaborate why the proposed layer-wise LR learns better than the previous work?
- Contrast the difference of the proposed method to the existing CBS.
- How much of empirical significance is there for choosing exponential based method?

---

> ### Author Response · Authors · 2023-11-20
> **Rebuttal letter**
>
> - _Marginal empirical gain. As shown in the Table 2 and 3, most of the gain over CBS, which is the direct competitors, is less than 1 or 2%._
>
> **Reply:** To determine if the reported accuracy gains observed for LeRaC with respect to the baseline are significant, we applied McNemar significance testing to the results reported in the main article on all 10 data sets. There are 26 test cases presented in Tables 2, 3 and 4 from the main paper. In 20 of 26 cases, we found that our results are significantly better than the corresponding baseline, at a p-value of 0.001. This confirms that our gains are statistically significant in the majority of cases. The results of the statistical tests were already included in the supplementary. Moreover, we underline that CBS leads to performance drops with respect to the conventional regime in 12 out 26 cases, while LeRaC improves the performance in each and every case. In other words, using CBS, one can expect performance drops in almost half of the test cases. In the experiments presented on 10 data sets, this does not happen with LeRaC. Hence, there is clear evidence showing that our improvements are not marginal, at least in about 50% of the cases, i.e. in the cases when CBS leads to performance drops.
>
> - _Method is too simple without intuitive ground that it should work better than others._
>
> **Reply:** In the supplementary, we provided a theoretical proof (Section 1)  and an empirical proof (Section 2) of our intuitive ground presented in the introduction. The reviewer raised many issues (including this one) that were already solved in the supplementary, which makes us believe that the reviewer did not read the supplementary material. Although we understand that reviewers have limited time to review, we kindly ask the reviewer to browse through the supplementary material, which strengthens our intuition presented in the main paper with comprehensive evidence.
>
> - _Although the analysis is intuitively sensible, the simplicity of the method brings marginal gain over CBS, even with the quite thorough ablation study. As authors mentioned, the empirically chosen exponential based method may not be the best choice or the intuition of noise amplification may not be a serious problem. Given the results, it is difficult to judge the main reason for the unsatisfactory performance._
>
> **Reply:** In the revised supplementary, we added new evidence showing that the SNR from one layer to the next drops exponentially (see Fig. 3 from the revised supplementary). This justifies why the exponential scheduler performs better. As mentioned above, we conducted statistical tests that clearly indicate that our performance gains are statistically significant, in the majority of cases (20 out of 26). Moreover, LeRaC surpasses CBS in 24 out of 26 test cases. we kindly ask the reviewer to read the supplementary, which contains convincing evidence against regarding the performance “unsatisfactory”.
>
> - _Can you elaborate why the proposed layer-wise LR learns better than the previous work?_
>
> **Reply:** We believe that both CBS and LeRaC share the same intuition and they are both simple methods. However, LeRaC is based on a more direct implementation, operating directly over the learning rates of neural layers, while CBS requires inserting additional architectural components, i.e. convolutions with Gaussian blur kernels. This reduces the applicability of CBS to neural architectures for which the application of blur kernels actually makes sense. The blur kernels also increase the training time of CBS (as per Fig. 1 from the supplementary). In summary, LeRaC has several advantages, which are demonstrated across a wide range of neural architectures and domains.
>
> - _Contrast the difference of the proposed method to the existing CBS._
>
> **Reply:** Unlike CBS, LeRaC does not introduce new operations, such as smoothing with Gaussian kernels, during training. As such, our approach does not increase the training time with respect to the conventional training regime, as shown in Fig. 1 from the supplementary. Another clear difference is the fact that LeRaC works for a wider variety of architectures and domains. This is confirmed by our experiments, where LeRaC brings performance gains across 26 test cases, while CBS leads to performance drops in 12 test cases.
>
> - _How much of empirical significance is there for choosing exponential based method?_
>
> **Reply:** As suggested, we performed additional statistical tests to compare the results of the exponential scheduler with the linear and the log schedulers. The tests show that the exponential is significantly better, at a p-value of 0.05. New results shown in Fig. 4 and 5 from the revised supplementary further support the exponential scheduler. Moreover, we added new evidence showing that the SNR from one layer to the next drops exponentially (see Fig. 3 in the revised supplementary). This further justifies why the exponential scheduler performs better.

---

> > ### Comment · Reviewer_RRAE · 2023-11-23
> > **Reviewer response**
> >
> > The reviewer sincerely thanks the authors for clarifications. The response clarifies my concerns of significance of the empirical results. And the reviewer appreciate the pointing out the empirical derivations in the supplementary material. But I still have the following concerns.
> > - For my concern about motivation or intuitive ground for the proposed method that is based on the empirical choice of log of exponents (in Eq.9) still persists. The design of the method (Eq.9) is still not intuitively or theoretically grounded but empirically chosen. Also, it may cause additional instability in computing learning rate as the equation involves exponent and log operations.
> > - For the comparison to CBS, I asked to clarify why it should performs better than CBS in terms of accuracy, not the other benefits.
> > - Additionally, many important arguments are in the supplementary materials. The reviewer encourages the authors to move some of the important conclusions to the main paper though we can leave the detailed derivations in the supplementary material (please note that ICLR allows appendix and usually the supplementary is for media other than texts).

---

> > > ### Author Response · Authors · 2023-11-23
> > > **Author reply**
> > >
> > > The authors thank the reviewer for reading the rebuttal and acknowledging our clarifications. We would like to clarify the remaining points below:
> > >
> > > - _For my concern about motivation or intuitive ground for the proposed method that is based on the empirical choice of log of exponents (in Eq.9) still persists. The design of the method (Eq.9) is still not intuitively or theoretically grounded but empirically chosen. Also, it may cause additional instability in computing learning rate as the equation involves exponent and log operations._
> > >
> > > **Reply:** Please note that Eq. 9 is actually very simple in practice, although the formula might seem complex. First, we note that $c^{log_c}$ cancels the exponent and the log. Moreover, in practice, learning rates are usually expressed as a power of $c=10$, e.g. $10^{-4}$. We provide two examples of applying Eq. 9 for $c=10$ below:
> > > 1. If we start with a learning rate of $10^{-8}$ for some layer $j$ and we want to increase it to $10^{-4}$ during the first $k=5$ epochs, then the intermediate learning rates are $10^{-7}$, $10^{-6}$ and $10^{-5}$, according to Eq. 9.
> > > 2. If we start with a learning rate of $10^{-8}$ for some layer $j$ and we want to increase it to $10^{-7}$ during the first $k=5$ epochs, then the intermediate learning rates are $10^{-7.75}$, $10^{-7.5}$ and $10^{-7.25}$, according to Eq. 9.
> > >
> > > In practice, notice that learning rates are expressed as one would expect, in terms of powers of ten. This aspect was already discussed in Section 4 (page 13) from the revised supplementary.
> > >
> > > - For the comparison to CBS, I asked to clarify why it should performs better than CBS in terms of accuracy, not the other benefits.
> > >
> > > **Reply:** Since CBS blurs the features in all layers, regardless of depth, it slows down the adaptation of all layers. In contrast, LeRaC prevents the adaptation of the final layers, while allowing the earlier layers to train as usual. In this sense, LeRaC is less aggressive, which could explain why it generally provides better accuracy rates than CBS.
> > >
> > > - _Additionally, many important arguments are in the supplementary materials. The reviewer encourages the authors to move some of the important conclusions to the main paper though we can leave the detailed derivations in the supplementary material (please note that ICLR allows appendix and usually the supplementary is for media other than texts)._
> > >
> > > **Reply:** Indeed, there are many experiments and discussion in the supplementary material. Since the supplementary material (15 pages) is almost twice as long as the main paper (9 pages), it was very difficult to decide what to include in the main paper and what to move to the supplementary. In the final version, we will move the images to the supplementary and bring more conclusions from the supplementary to the main paper, as suggested by the reviewer.

---

### Official Review · Reviewer_3pG7 · 2023-10-30

**Soundness:** 3 good
**Presentation:** 2 fair
**Contribution:** 1 poor
**Rating:** 5
**Confidence:** 4

**Summary:**

This paper proposes a learning rate curriculum (LeRaC) approach for the effective training of deep networks. Specifically, LeRaC assigns higher learning rates to neural layers closer to the inputs, gradually decreasing the learning rates as the layers are placed farther away
from the inputs. The learning rates increase at various paces during the first training iterations, until they all reach the same value. Empirical results on top of images, language, and audio are provided. LeRaC outperforms CBS.

**Strengths:**

1. The proposed method is simple and easy to implement.
2. The experiments in the paper are extensive.

**Weaknesses:**

1. My major concern lies on that, it is difficult to understand why LeRaC is effective. The motivation is questionable. The paper says that a random parameter initialization results in a propagation of noise. It seems that this issue can be well addressed with the widely-used standard warm-up strategy. LeRaC seems to be only an incremental contribution on top of the most common case.

2. More  theoretical analysis on the effectiveness of LeRaC will make this paper more convincing.

3. The results on full-ImageNet are absent, which I think is necessary.

4. The authors may consider citing [*1-*4] and comparing with them.

[*1] Zhou, Tianyi, and Jeff Bilmes. "Minimax curriculum learning: Machine teaching with desirable difficulties and scheduled diversity." International conference on learning representations. 2018.

[*2] Zhou, Tianyi, Shengjie Wang, and Jeffrey Bilmes. "Curriculum learning by dynamic instance hardness." Advances in Neural Information Processing Systems 33 (2020): 8602-8613.

[*3] Dogan, Ürün, et al. "Label-similarity curriculum learning." Computer Vision–ECCV 2020: 16th European Conference, Glasgow, UK, August 23–28, 2020, Proceedings, Part XXIX 16. Springer International Publishing, 2020.

[*4] Wang, Yulin, et al. "Efficienttrain: Exploring generalized curriculum learning for training visual backbones." Proceedings of the IEEE/CVF International Conference on Computer Vision. 2023.



**Post-rebuttal**

Thank you for the response from the reviewers. Although some of my concerns are solved, I'm still leaning towards rejection. However, I'm happy to raise my score to "5: marginally below the acceptance threshold".

My major concern that remains unsolved is that, personally, I think a more comprehensive evaluation on full-ImageNet is necessary for the current deep learning community to accept such an "empirical-oriented"  method.

**Questions:**

See weaknesses.

---

> ### Author Response · Authors · 2023-11-20
> **Rebuttal letter**
>
> - _It is difficult to understand why LeRaC is effective. The motivation is questionable. The random parameter initialization resulting in a propagation of noise can be well addressed with a standard warm-up strategy._
>
> **Reply:** In the supplementary, we provide a theoretical proof (Section 1)  and an empirical proof (Section 2) of our motivation, i.e. the fact that “a random parameter initialization results in a propagation of noise”. It is therefore not clear why our motivation is “questionable”. If the reviewer considers that the theoretical proof is wrong, we kindly ask the reviewer to be more specific and indicate the mistake. If the reviewer considers that the empirical results are not credible, we kindly ask the reviewer to try out the code provided with the supplementary. How can we further convince the reviewer, aside from giving a theoretical and an empirical proof?
> Regarding the warm-up comment, we underline that all the CvT-13 results based on the “conventional” regime reported in Table 2 use Linear Warmup with Cosine Annealing, this being the recommended scheduler for CvT (Wu et al., 2021). When we introduce LeRaC, we simply deactivate Linear Warmup with Cosine Annealing between epochs 0 and k. Hence, our experiments already include a direct comparison between a widely-used warm-up strategy and LeRaC. The results show that LeRaC leads to significantly better results, according to McNemar testing at a p-value of 0.001. This aspect was too already clarified in the supplementary.
>
> - _More theoretical analysis on the effectiveness of LeRaC will make this paper more convincing._
>
> **Reply:** We provided a theoretical proof of the underlying assumption of our method in the supplementary. In general, the reviewer raised many issues that were already solved in the supplementary, which makes us believe that the reviewer did not read the supplementary material. Aside from the theoretical proof (Section 1 from the supplementary), we also present several experiments to empirically demonstrate the underlying hypothesis / motivation of LeRaC in Section 2 from the supplementary. Moreover, the comprehensive evaluation on 10 data sets from different domains based on a large variety of neural architectures clearly indicate that LeRaC is consistent in surpassing the standard training regime, as well as other curriculum-based methods. Although we understand that reviewers have limited time to review, we kindly ask the reviewer to browse through the supplementary material. We strongly believe that the (revised) supplementary material addresses all the concerns raised by the reviewer.
>
> - _The results on full-ImageNet are absent, which I think is necessary._
>
> **Reply:** We have limited computational resources and we considered that it is more important to present results on 10 data sets from multiple domains (vision, language, audio) and test multiple architectures. Hence, we sampled 200 classes from ImageNet-1K and performed the experiments on this subset (see results on ImageNet-200 shown in Table 2). Moreover, we observe that not all the references indicated by the reviewer, e.g. [*1], report results on ImageNet, confirming that full ImageNet experiments are not always a necessity. Still, in the limited time given for the rebuttal, we were able to run ResNet-18 based on LeRaC on the full ImageNet. The results are shown in the newly added Table 10 from the updated supplementary. Our improvements on ImageNet-1K over CBS and [*4] are close to 1%, further confirming the consistency of LeRaC, which we already observed on 10 data sets from multiple domains. We thank the reviewer for this suggestion, which greatly enhances our empirical validation.
>
> - _The authors may consider citing [*1-*4] and comparing with them._
>
> **Reply:** First, please note that we already have a comparison with [*3] in the supplementary (see Table 7 from the supplementary), showing that our method achieves superior results. Moreover, we note that [*4] should be regarded as concurrent work, since ICCV 2023 took place after the submission deadline for ICLR 2024. Nevertheless, we compare with [*4] in the newly added Tables 10 and 11 from the updated supplementary. Table 10 shows that our improvement on ImageNet-1K over [*4] for ResNet-18 is 0.96%. Table 11 indicates that we obtain similar improvements on CIFAR-10 and CIFAR-100, where LeRaC surpasses EfficientTrain [*4] in 3 out of 4 cases. In summary, the added results are favorable to LeRaC. We also note that the code for [*1] is not publicly available. We asked the authors to share the code, but they were not able to share it because it depends on a private package. The code for [*2] is available, but it does not seem to work, i.e. the loss becomes NaN after a few epochs, in spite of our efforts to repair the code and make it work. We thank the reviewer for this recommendation, which strengthens our empirical validation.

---

### Official Review · Reviewer_i4Pq · 2023-11-01

**Soundness:** 3 good
**Presentation:** 3 good
**Contribution:** 2 fair
**Rating:** 5
**Confidence:** 3

**Summary:**

A per-layer learning rate schedule that assigns a higher learning rate for initial layers in the initial epochs then equalises the learning rate among all layers throughout the remaining learning process.

**Strengths:**

- Formulating Curriculum Learning as a learning rate scheduling problem is not a contribution of this paper but the exposition provided here presents a good argument for this.
- It is empirically shown that LeRaC achieves better performance than baselines over a wide range of architectures and tasks.
- The paper was clear and easy to follow albeit attempted to over-complicate matters in certain areas (e.g., the first 2-3 paragraphs of Section 3).

**Weaknesses:**

- While the experiments focus on architectures and tasks and some ablation studies, no analysis is provided to empirically demonstrate the claims in the paper. For example, no learning curves were presented in the paper (some learning curves were presented in the supplementary material but only compared to CBS) and similarly, no activation maps (some presented in the supplementary material but only compared to conventional training). A convincing argument must be presented that does not only focus on the final performance but demonstrates properties of the learning process compared against a number of learning rates, schedules, baselines.
- The interplay between this learning rate scheduler and initialisation/optimisers has not been studied.

**Questions:**

- Can the improvement in learning dynamics be demonstrated empirically in a wide variety of settings as critiqued above?
- How does LeRaC interface with different initialisers/optimisers?

---

> ### Author Response · Authors · 2023-11-20
> **Rebuttal letter**
>
> - _The experiments focus on architectures and tasks and some ablation studies, with little analysis provided to empirically demonstrate the claims in the paper. A convincing argument to demonstrate the properties of the learning process compared against a number of learning rates, schedules, etc, must be provided._
>
> **Reply:** We thank the reviewer for reading the supplementary material. As recommended, we included additional graphs in the revised supplementary to show the effect of various learning rates and schedulers on the learning progress. More specifically, we present results with various schedulers in Figures 4 and 5 from the revised supplementary, and results with various learning rates in Figures 6, 7 and 8. The new results further confirm the benefits of the exponential scheduler and the use of gradually decreasing initial learning rates. We thank the reviewer for this observation, which improves our validation.
>
> - _The interplay between this learning rate scheduler and initialisation/optimisers has not been studied._
>
> **Reply:** Indeed, this point was not well highlighted in the paper. We employed LeRaC on a high variety of architectures, using the most suitable optimizer for each architecture. Hence, LeRaC was tested in combination with SGD, Adam, AdamW and AdaMax (as per Table 1). LeRaC brings performance improvements, regardless of the selected optimizer. This demonstrates that LeRaC is compatible with a variety of widely-used optimizers. Regarding initialization, please note that some of the architectures are trained from scratch (and use Glorot init), while others come with pre-trained weights (e.g. CvT-13 pre-trained, BERT pre-trained). Once again, LeRaC brings performance improvements, regardless of the weight initialization. This confirms that LeRaC is suitable for both fine-tuning and training from scratch. Moreover, we underline that all the CvT-13 results based on the “conventional” regime reported in Table 2 use Linear Warmup with Cosine Annealing, this being the recommended scheduler for CvT (Wu et al., 2021). When we introduce LeRaC, we simply deactivate Linear Warmup with Cosine Annealing between epochs 0 and k. Hence, our experiments already include a direct comparison between a widely-used warm-up strategy and LeRaC. The results show that LeRaC leads to significantly better results, according to McNemar testing at a p-value of 0.001. This indicates that LeRaC can successfully replace widely-used initialization schemes.
>
> - _Can the improvement in learning dynamics be demonstrated empirically in a wide variety of settings as critiqued above? How does LeRaC interface with different initialisers/optimisers?_
>
> **Reply:** The questions correspond to the identified weak points, which we already addressed above.

---

> > ### Comment · Reviewer_i4Pq · 2023-11-23
> > **Maintain Rating**
> >
> > I thank the authors for the rebuttal. After considering the rebuttal and comments made by other reviewers, I have decided to maintain my rating. While new results have been added in this discussion phase, further validation is required for a complete and convincing argument.

---

> > > ### Author Response · Authors · 2023-11-23
> > > **Comment on validation**
> > >
> > > Thank you for considering our rebuttal. We would like to address the remaining concern below
> > >
> > > - _Further validation is required for a complete and convincing argument._
> > >
> > > **Reply:** We kindly emphasize that our validation is more comprehensive than several similar studies, for example:
> > > 1. Curriculum by Smoothing (CBS) (NeurIPS 2020) is validated on **8** datasets.
> > > 2. [*1] (ICLR 2018) is validated on **7** datasets.
> > > 3. [*3] (ECCV 2020) is validated on **5** datasets.
> > > 4. [*4] (ICCV 2023) is validated on **6** datasets.
> > >
> > > In contrast, our method is now validated on **11** datasets. Therefore, it is not clear why our method requires further validation than these related studies.
> > >
> > > [*1] Zhou, Tianyi, and Jeff Bilmes. "Minimax curriculum learning: Machine teaching with desirable difficulties and scheduled diversity." International conference on learning representations. 2018.
> > >
> > > [*3] Dogan, Ürün, et al. "Label-similarity curriculum learning." Computer Vision–ECCV 2020: 16th European Conference, Glasgow, UK, August 23–28, 2020, Proceedings, Part XXIX 16. Springer International Publishing, 2020.
> > >
> > > [*4] Wang, Yulin, et al. "Efficienttrain: Exploring generalized curriculum learning for training visual backbones." Proceedings of the IEEE/CVF International Conference on Computer Vision. 2023.

---

### Meta-Review · Area_Chair_JBXq · 2023-12-06

**Metareview:**

The paper proposes a new curriculum learning method that uses different learning rates for each layer of the network. A comprehensive set of experiments across different tasks and architectures showcases the effectiveness of the proposed method. The reviewers agreed that the proposed curriculum method is easy to implement and can be integrated into a wide range of architectures. However, the reviewers raised a common concern that the paper needs deeper empirical or theoretical analysis to gain better insights into the effectiveness and workings of the proposed curriculum method. We want to thank the authors for their detailed responses. Based on the raised concerns and follow-up discussions, unfortunately, the final decision is a rejection. Nevertheless, this is exciting and potentially impactful work, and we encourage the authors to incorporate the reviewers' feedback when preparing a future revision of the paper.

**Justification For Why Not Higher Score:**

The reviewers have borderline ratings and are inclined toward a rejection decision. As per the reviews and discussions, the paper would benefit from a deeper empirical or theoretical analysis to gain better insights into the effectiveness and workings of the proposed curriculum method.

**Justification For Why Not Lower Score:**

N/A

---

### Decision · Program_Chairs · 2024-01-16

Reject